# Mixture-of-Linguistic-Experts Adapters for Improving and Interpreting Pre-trained Language Models

**Raymond Li[†], Gabriel Murray[‡], Giuseppe Carenini[†]**
[†] University of British Columbia, Vancouver, BC, Canada
[‡] University of Fraser Valley, Abbotsford, BC, Canada
{raymondl, carenini}@cs.ubc.ca
gabriel.murray@ufv.ca

## Abstract

In this work, we propose a method that combines two popular research areas by injecting linguistic structures into pre-trained language models in the parameter-efficient fine-tuning (PEFT) setting. In our approach, parallel adapter modules encoding different linguistic structures are combined using a novel Mixture-of-Linguistic-Experts architecture, where Gumbel-Softmax gates are used to determine the importance of these modules at each layer of the model. To reduce the number of parameters, we first train the model for a fixed small number of steps before pruning the experts based on their importance scores. Our experiment results with three different pre-trained models show that our approach can outperform state-of-the-art PEFT methods with a comparable number of parameters. In addition, we provide additional analysis to examine the experts selected by each model at each layer to provide insights for future studies.

## 1 Introduction

In recent years, pre-trained language models have become the de facto instrument for the field of natural language processing (NLP) (Devlin et al., 2019; Liu et al., 2019; Clark et al., 2020; He et al., 2021, 2023). This shift is largely due to the emergence and success of transformer-based models (Vaswani et al., 2017) where large-scale pre-training helps the model to learn the syntactic and semantic structure of a language without explicit supervision. At the same time, there are good reasons to question whether these models can be said to understand a language in any meaningful and interpretable way (Trott et al., 2020; Merrill et al., 2021). To address this conundrum, probing studies have demonstrated, to a certain extent, that it is possible to infer linguistic structures from the representations within these models (Hewitt and Manning, 2019; Tenney et al., 2019b; Maudslay et al., 2020). However, the precise connection between the existence of structures and their benefits to task performance is yet to be firmly established. On the other hand, while the conventional way of fine-tuning has found success in a wide array of NLP tasks, its applicability has increasingly diminished due to the associated computational expense with the recent shift towards larger and more complex models (Zhao et al., 2023).

While some have argued that pre-training on unstructured text alone equips the model with sufficient capacity to comprehend the meaning of language, others (Bender and Koller, 2020; Prange et al., 2022) have asserted that mapping the model's behavior onto human-comprehensible structures offers more dependable evidence of its ability to tackle tasks beyond merely exploiting superficial cues. Specifically, studies in this area have yielded successful attempts to inject syntactic and semantic structures into the pre-trained language models (Bai et al., 2021; Wu et al., 2021; Yu et al., 2022), with positive results reported on downstream tasks. However, despite recent efforts, no existing work has addressed the problem of where and how to effectively inject multiple different structures in an efficient manner.

The conventional approach of fine-tuning pre-trained NLP models involves optimizing the full set of model parameters for each task. However, this results in a separate copy of fine-tuned model parameters for each task and has become increasingly infeasible due to the recent trend of pre-training larger and larger models. To address these concerns, a surge of recent work has been dedicated to the study of parameter-efficient fine-tuning (PEFT) methods (Ding et al., 2023), where only a small portion of task-specific trainable parameters are tuned while keeping the rest of the model frozen. While these studies have achieved impressive performance even comparable to the full fine-tuning, they have been mostly focused on either determining the subset of model parameters for tuning (Lee

et al., 2019; Ben Zaken et al., 2022) or finding the location to insert additional trainable parameters (Houlsby et al., 2019a; Li and Liang, 2021; Hu et al., 2022). No existing work has addressed the problem of whether linguistic structural priors can be incorporated into these trainable parameters under the PEFT setting.

In this work, we align the two research areas of injecting linguistic structures and PEFT by proposing a strategy of effectively combining multiple linguistic structures into pre-trained NLP models in a parameter-efficient fashion. To combine multiple linguistic structures, we propose a novel architecture inspired by Mixture-of-Experts models (Shazeer et al., 2017), where Relational Graph Convolutional Networks (RGCN) modules (Schlichtkrull et al., 2018) encoded with different linguistic trees are aggregated using learnable Gumbel-Softmax (Jang et al., 2017) gates, and inserted between each layer of the pre-trained model. To reduce the number of parameters, we propose a pruning strategy where we first tune the full set of RGCN modules before pruning all but the top "experts" based on the importance score learned from the gates. To demonstrate the benefits of our approach, we perform experiments on the GLUE benchmark with three different pre-trained NLP models and compare the results with state-of-the-art PEFT methods (Mao et al., 2022). Further, we perform additional analysis to understand which types of linguistic structures are kept at each layer of the model and provide insights for future work on injecting knowledge through PEFT methods. In short, our contributions can be summarized as the following:

1. We propose a novel architecture to effectively combine and interpret multiple linguistic structures at different layers of the pre-trained model.

2. To improve efficiency, we adopt a pruning strategy by keeping only the top experts according to their importance scores.

3. Our experimental results with three different models demonstrate the benefits of our approach by achieving the best overall performance on the GLUE benchmark.

4. We perform analysis on the experts selected by the model to providing valuable insights for future work.

## 2 Related Works

We organize this section based on the two research areas that our work seeks to align. In §2.1, we provide an overview of techniques to inject linguistic structure, while §2.2 summarizes recent trends in parameter-efficient fine-tuning.

### 2.1 Injecting Linguistic Structures

Earlier works on injecting linguistic structures into neural networks are often based on the recursive neural network architecture (Goller and Kuchler, 1996; Socher et al., 2011, 2012, 2013), where a compositional function recursively combines representations of child nodes following a predefined tree structure. Following the same intuition, subsequent studies have extended their approach for composing hidden states into a variety of neural architectures including recurrent neural networks (RNNs) (Tai et al., 2015; Miwa and Bansal, 2016; Roth and Lapata, 2016; Kuncoro et al., 2017; Shen et al., 2019), graph neural networks (GNNs) (Marcheggiani and Titov, 2017; Bastings et al., 2017; Zhang et al., 2018; Huang and Carley, 2019; Wang et al., 2020), and later, Transformers (Wu et al., 2018; Hao et al., 2019; Strubell et al., 2018; Wang et al., 2019b,c). For instance, Strubell et al. (2018) used the bi-affine operator (Dozat and Manning, 2017) to predict the affinity score between the token representations (key and query vector) based on the dependency tree, while (Wang et al., 2019c) encouraged the attention heads to follow tree structures by applying a constituent prior on the attention weights.

More recently, research in this area has shifted towards pre-trained language models (Devlin et al., 2019; Liu et al., 2019; Clark et al., 2020; He et al., 2021, 2023). While prior studies on probing (Hewitt and Manning, 2019; Tenney et al., 2019b; Maudslay et al., 2020; Newman et al., 2021; Arps et al., 2022) have shown that meaningful hierarchical structures (e.g., syntactic trees) can be extracted from pre-trained models without explicit supervision, it has also been found that incorporating linguistic structures can still be beneficial for downstream performance (Zhang et al., 2020; Kuncoro et al., 2020; Sachan et al., 2021; Qian et al., 2021), even when the structures already exist in the model (Li et al., 2022). For example, Bai et al. (2021) explicitly masked the existing pre-trained attention weights based on the adjacency matrices defined by the syntactic trees. On the other

hand, Wu et al. (2021) used additional GNN layers to incorporate semantic dependencies by appending them on top of the pre-trained encoder. Most similar to our work, Yu et al. (2022) extended the approach by Wu et al. (2021) and performed an empirical study on syntax and trivial graphs. However, their method requires training a new model for each graph, which is inefficient for studying their benefits at different layers of the model. To the best of our knowledge, no existing works have attempted to incorporate multiple different linguistic structures within the same model, as we do in this paper.

## 2.2 Parameter Efficient Fine-tuning

While the standard paradigm of fine-tuning pre-trained language models has emerged as a common practice for NLP tasks (Min et al., 2021), it has become less applicable due to the computational cost associated with the increasingly large models (Brown et al., 2020a; OpenAI, 2023). Parameter-efficient fine-tuning (PEFT) methods (Ding et al., 2023), on the other hand, present a solution to this problem by freezing most or all of the pre-trained weights and only fine-tuning a small set of parameters in proportion to the model size. PEFT methods can be roughly organized into two categories. The first category tunes a subset of existing parameters with notable examples including freezing entire layers (Lee et al., 2019) or tuning only the bias terms (Ben Zaken et al., 2022). However, these approaches generally lead to worse performance and have only been shown to achieve comparable performance to full fine-tuning on low-resource tasks. Alternatively, the second category adds new trainable parameters while keeping the pre-trained weights frozen (Han et al., 2021; Karimi Mahabadi et al., 2021; Lester et al., 2021). For example, Houlsby et al. (2019a) used a trainable bottleneck layer after the feed-forward network in each layer of the model, Li and Liang (2021) prepended trainable vectors to the input of multi-head attention, while Hu et al. (2022) combined the pre-trained attention weights with trainable low-rank matrices. Lastly, more recent studies (He et al., 2022; Mao et al., 2022) proposed a unified framework by combining different PEFT methods as sub-modules. While we use their approach as our baselines, no existing PEFT works have attempted to incorporate interpretable structures as priors to the trainable modules, as we do in this paper.

## 3 Model Architecture

In this section, we describe the architectures of our Mixture-of-Linguistic adapters (Figure 1). We start by first introducing the Relational Graph Convolutional Network (RGCN) modules for incorporating linguistic structures (§3.1) before describing the method used for combining multiple RGCN (§3.2). Finally, we discuss how the adapters are inserted into the pre-trained model (§3.3).

## 3.1 Modeling Dependency Structures

To model dependency structures, we adopt the method proposed by Wu et al. (2021), where RGCN (Schlichtkrull et al., 2018) layers are used to propagate node representations according to the structure defined by the dependency tree.

$$h_i^{(\ell)} = \text{ReLU}\left( \sum_{r \in \mathcal{R}} \sum_{j \in \mathcal{N}_i} \frac{W_r h_j^{(\ell-1)}}{|\mathcal{N}_i|} + W_0 h_i^{(\ell-1)} \right) \tag{1}$$

Equation 1 describe the propagation process for a single RGCN layer, where the node representation $h_i$ is updated with a learned composition function based on the node's neighbors $h_j \in \mathcal{N}_i$ (and itself) in the dependency graph. Specifically, we use the intermediate hidden states of the pre-trained model as input, where sub-word token vectors are mean-pooled to create the node representation for the associated word. Since the number of parameters in RGCN linearly increases with the number of relation types, rather than associating each dependency relation with a separate set of weights $W_r$, we only model the child and parent relations ($|\mathcal{R}| = 2$) to reduce parameter count.

The graph convolution operation has a computational complexity $\mathcal{O}(|E| \cdot d_1 \cdot d_2)$, where $d_1$ and $d_2$ are respectively the number of input and output dimensions of the layer, and $|E|$ is the total number of edges defined by the dependency graph. In addition, the self-loop operation in the RGCN layer adds a complexity $\mathcal{O}(|N| \cdot d_1 \cdot d_2)$, where $|N| = |E| + 1$ is the total number of nodes or word tokens in the dependency graph. The self-loop operation has the same complexity as the standard linear layer.

## 3.2 Combining Different Modules

Inspired by the Mixture-of-Experts architecture (Shazeer et al., 2017), we propose a strategy

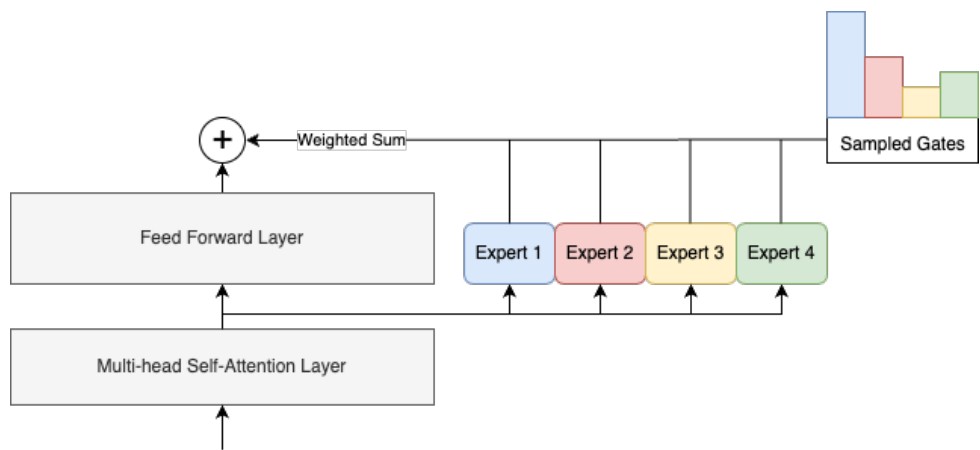

Figure 1: Our proposed Mixture-of-Linguistic-Experts architecture for a single transformer layer. In the four-expert configuration provided by the example, where the outputs from the expert modules are aggregated based on the weights sampled from the Gumbel-Softmax distribution.

to determine the importance of different adapter modules by sampling gate values from a Gumbel-Softmax distribution (Maddison et al., 2017; Jang et al., 2017). Specifically, we define a gate logit $z_i$ for each "expert" module $E_i$, where the gate value $g_i$ is sampled from the Gumbel-Softmax distribution during training. The sampling method is defined as:

$$g_i = \text{softmax}(z_i + \epsilon)/\tau \quad (2)$$

where the stochasticity comes from the gumbel noise $\epsilon = -\log(-\log(u))$ s.t. $u \sim \text{Uniform}(0, 1)$, and $\tau$ is the temperature to control the randomness of the distribution. The value of the gate logit $z_i$ can be interpreted as the contribution of the respective expert module when computing the aggregated representation from all experts.

In contrast to the softmax gates used in the original MoE architecture (Shazeer et al., 2017), sampling gates from the Gumbel-Softmax distribution provides more interpretability regarding its importance since the inherent stochasticity introduces an exploratory characteristic and allows the model to consider a diverse set of potential outputs. Meanwhile, the standard softmax operation assumes a single correct answer at each layer, meaning the model could possibly overlook good combinations of modules by locally exploiting the module with the single highest probability.

### 3.3 Adapters

Based on prior works on parameter efficient fine-tuning (Houlsby et al., 2019b; Mao et al., 2022), we inject our Mixture-Linguistic-Experts layer between the layers of pre-trained transformer model

(§3.2) and update only the adapters while keeping the pre-trained parameters frozen. We choose to insert modules following the suggestions by He et al. (2022), where they found that inserting adapters in parallel to the feed-forward networks (FFN) achieves the overall best performance.

$$h_{\text{attn}}^{(\ell)} = \text{MultiHeadAttn}(h^{(\ell-1)})$$
$$h^{(\ell)} = \text{FFN}(h_{\text{attn}}^{(\ell)}) + \text{Adapter}(h_{\text{attn}}^{(\ell)}) \quad (3)$$

From Equation 3, our adapter module takes the hidden output $h_{\text{attn}}$ from the multi-head attention (MultiHeadAttn) sub-layer and uses an additive composition with the original FFN to create the final layer output $h^{(l)}$.

## 4 Training Strategy

While the architecture proposed in section 3 allows us to aggregate multiple adapter modules at each pre-trained layer, it significantly decreases the efficiency due to the number of task-specific parameters used during training and inference. To address this issue, we propose a pruning strategy to reduce the number of experts.

In order to decide which expert to keep at each layer, we first fine-tune the full set of expert modules using our Mixture-of-Linguistic-Experts architecture (Figure 1) for a fixed small number of steps. After the gates have converged, importance score from the gates can be used to determine which experts to keep. While an iterative pruning strategy (Michel et al., 2019; Behnke and Heafield, 2020; Tan and Motani, 2020) can also be used, it is less efficient due to the requirement of more training steps. Finally, after the pruning process, we restart the

training process and fine-tune the resulting model with one expert module per layer.

# 5 Experiments

We describe in detail the experimental settings and results in this section. We start by providing a brief summary of the linguistic graphs (§5.1) before describing the datasets (§5.2) models (§5.3), and the hyperparameters settings (§5.4). Finally, we present the results in §5.5.

## 5.1 Linguistic Graphs

In our experiments, we use three different linguistic graphs to encode sentence-level structures. Following prior studies (Wu et al., 2021; Yu et al., 2022), we infuse the semantic and syntactic dependency trees as well as a sequential bidirectional graph into three separate RGCN adapter modules for each layer of the pre-trained model. In addition, to account for scenarios where structures are either not needed or harmful, we also use a multi-layer perception (MLP) module to represent an edgeless graph, where no composition is performed.

**Syntactic Trees** In syntactic parses, each word in the sentence is assigned a syntactic head based on Universal Dependencies (UD) formalism (de Marneffe et al., 2021). We use the Bi-LSTM-based deep biaffine neural dependency parser (Dozat and Manning, 2017) trained on the English UD treebank from the Stanza library (Qi et al., 2020).

**Semantic Trees** Based on the DELPH-IN dependencies formalism (Ivanova et al., 2012), semantic parses assign word dependencies based on predicate-argument relations. In contrast to syntactic graphs, words that do not contribute to the meaning representation of the sentence do not appear in the semantic graph. The graphs are extracted with a neural transition-based parser (Wang et al., 2018; Che et al., 2019) trained on the CoNLL 2019 shared task (Oepen et al., 2019).

**Sequential Bidirectional Graphs** We also use a straight-forward sequential bidirectional graph that connects word tokens in a sequential order. This allows the RGCN layers to aggregate local information rather than potentially long dependencies, where it has shown the ability to improve task performance when injected into pre-trained transformer layers via fixed attention (Li et al., 2022).

**Edgeless Graphs** In addition to the three linguistic graphs, we also apply a straight-forward nonlinear transformation using MLP layers. The intuition is that at some layers, injecting structures might be unhelpful (or even detrimental) to the task performance when the linguistic prior cannot be utilized based on the representation learned by that layer.

## 5.2 Datasets

| Dataset | Task | Train | Dev |
|---------|------|-------|-----|
| CoLA | Acceptability | 1K | 1.74 |
| RTE | Entailment | 2.5K | 278 |
| MRPC | Paraphrase | 2.7K | 409 |
| STS-B | Similarity | 5.8K | 1.5k |
| SST-2 | Sentiment | 67K | 873 |
| QNLI | Entailment | 105k | 5.5K |
| QQP | Entailment | 363K | 40K |
| MNLI | Entailment | 392k | 9.8K |

Table 1: The statistics of the datasets in the GLUE benchmark, ordered by the size of the training set.

We conduct all our experiments on the GLUE benchmark (Wang et al., 2019a), consisting of a comprehensive suite of natural language understanding tasks. The benchmark contains eight datasets for text classification, including linguistic acceptability (CoLA), sentiment analysis (SST-2), similarity and paraphrase tasks (MRPC, STS-B, QQP), and natural language inference (MNLI, QNLI, RTE). For evaluation metric, we use Matthew's Correlation for CoLA, F1 for MRPC and QQP, Spearman's Rank-Order Correlation for STS-B, and Accuracy for SST-2, RTE, QNLI, and MNLI. Following prior studies (Houlsby et al., 2019b; He et al., 2022), we exclude the WNLI dataset from our experiments due to its limited coverage. The statistics of the datasets are presented in Table 1.

## 5.3 Models

In our experiments, we apply our methods to three different pre-trained language models: BERT, RoBERTa, DeBERTaV3. RoBERTa (Liu et al., 2019) enhances BERT (Devlin et al., 2019) by incorporating more training data and removing the next-sequence prediction objective, DeBERTa (He et al., 2021) introduced a disentangled attention mechanism for encoding relative positions at every layer, while DeBERTaV3 (He et al., 2021) improved upon the prior versions by adapting the

| Method | CoLA | RTE | MRPC | STS-B | SST-2 | QNLI | QQP | MNLI | Average |
|---|---|---|---|---|---|---|---|---|---|
| BERT | | | | | | | | | |
| Full Fine-Tuning | 62.08 | 66.43 | 90.94 | 89.76 | 91.63 | 89.95 | 87.35 | 83.23 | 82.67 |
| Adapter | 61.51 | 71.84 | 89.86 | 88.63 | 91.86 | 90.55 | 86.78 | 83.14 | 83.02 |
| Prefix-tuning | 55.37 | 76.90 | 91.29 | 87.19 | 90.94 | 90.39 | 83.30 | 81.15 | 82.07 |
| LoRA | 60.47 | 71.48 | 90.03 | 85.65 | 91.51 | 89.93 | 85.98 | 82.51 | 82.20 |
| UniPELT (AP) | 61.15 | 71.84 | 90.28 | 88.86 | 91.86 | 90.77 | 86.74 | 83.41 | 83.12 |
| UniPELT (APL) | 61.53 | 73.65 | 90.94 | 88.93 | 91.51 | 90.50 | 87.12 | 83.89 | 83.51 |
| Ours | 61.49 | 70.36 | 90.43 | 88.71 | **92.66** | **93.03** | **87.82** | **84.30** | 83.60 |
| RoBERTa | | | | | | | | | |
| Full Fine-Tuning | 68.0 | 86.6 | 90.9 | 92.4 | 96.4 | 94.7 | 92.2 | 90.2 | 88.9 |
| UniPELT (APL) | 61.91 | 74.31 | 91.74 | 90.26 | 93.92 | 92.00 | 87.68 | 87.23 | 84.88 |
| Ours | 62.20 | 72.32 | 92.77 | 90.34 | **94.27** | **92.53** | **88.29** | **87.83** | 85.07 |
| DeBERTaV3 | | | | | | | | | |
| Full Fine-Tuning | - | - | - | - | - | - | 90.7 | - | - |
| UniPELT (APL) | 68.02 | 81.59 | 92.42 | 91.70 | 95.64 | 93.65 | 89.60 | 89.13 | 87.72 |
| Ours | 69.93 | 79.42 | 93.38 | 91.01 | 95.84 | 93.92 | **89.83** | **89.47** | 87.85 |

Table 2: Results on the GLUE benchmark for BERT, RoBERTa, and DeBERTaV3. For BERT, the full fine-tuning and PEFT baseline results are directly copied from Mao et al. (2022). For both RoBERTa and DeBERTaV3, the UniPELT results are obtained using the AdapterHub implementation (Pfeiffer et al., 2020), while the full fine-tuning results are copied from their original papers (RoBERTa only reported precision to the tenth decimal place, while DeBERTaV3 only reported the base model on QQP). All our results are averages over three seeds, with statistically significant improvements (>99% confidence Bootstrap Test) over UniPELT highlighted in **bold**.

replaced token detection objective (Clark et al., 2020). For all models, we use the standard variant with 12 layers and 12 heads. For baselines, we use the unified framework for parameter-efficient language model tuning (UniPELT) proposed by (Mao et al., 2022). Since the results from the original paper demonstrated superior performance over other PEFT methods (Houlsby et al., 2019a; Li and Liang, 2021; Hu et al., 2022), we only report the results for these methods for BERT. For all tasks, we apply a classifier on the [CLS] token representation from the last hidden layer.

### 5.4 Hyperparameters

Both MLP and RGCN adapter modules consist of two hidden layers with a bottleneck dimension of 48. Since RGCN modules require $3\times$ the number of parameters as MLP modules, we only select the top-2 RGCN modules based on their gate values. Following the settings by Mao et al. (2022), we set the input length to 128, and train for a total of 50 epochs for with a learning rate of $5e-4$ and batch size of 16. During the initial steps of training our Mixture-of-Linguistic-Experts model, we follow the suggestions from prior work (Huijben et al., 2022) and apply temperature annealing (Jang et al., 2017) to gradually decrease the temperature

from 5 to 0.1 over 1000 steps. The intuition behind temperature annealing is to allow the model to start with a more exploratory behavior before gradually becoming more exploitative. Lastly, we also scale the adapter output by a constant factor of 4 as proposed in the work by He et al. (2022).

### 5.5 Results

From the results in Table 2, we see that our approach achieves the best overall performance on the GLUE benchmark. For individual tasks, although our method lags behind UniPELT in the low-resource tasks of RTE and STS-B, our method achieves consistent improvements in the four tasks with the highest number of training examples (Table 1): SST-2, QNLI, QQP and MNLI, where the improvements for SST-2 and QNLI are statistically significant for two out of the three models, and QQP and MNLI for all three models. This is consistent with the findings by prior work (Mao et al., 2022; Chen et al., 2022), where they found that while existing PEFT methods excel in low-resource tasks, they still struggle to yield consistently competitive performance in medium and high-resource settings. However, we believe that learning to use the linguistic structures associated with the dependency trees requires more tuning and can outper-

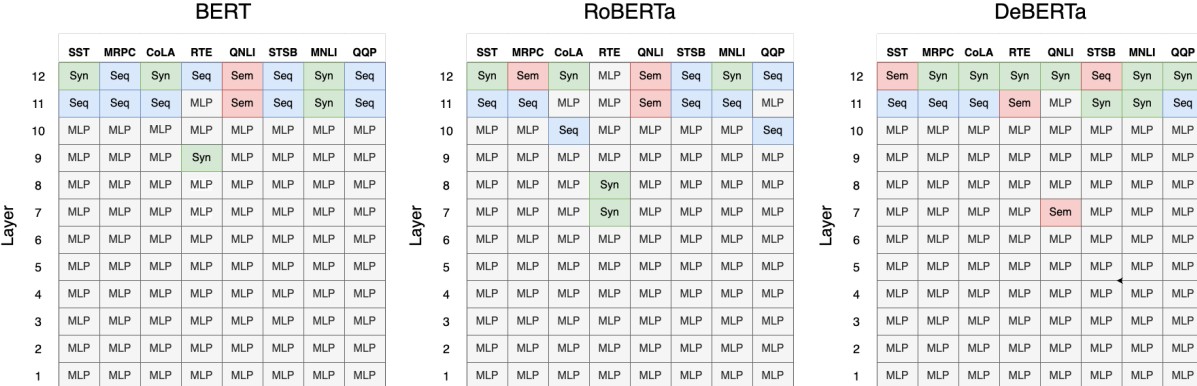

Figure 2: These heatmaps illustrate which of the four expert modules are used at each layer for the three models. We use green, red, blue, and grey to represent the syntactic (Syn), semantic (Sem), sequential (Seq), and MLP expert, respectively.

form standard PEFT methods when there are more training data available. Lastly, it is worth highlighting that the application of our approach to the RoBERTa model resulted in a notable increase in performance (+0.19) over the baseline, surpassing the gains observed with BERT (+0.09) and DeBERTa (+0.13). Since RoBERTa is pre-trained on a larger corpus than BERT, we hypothesize that this discrepancy could be due to the fact that RoBERTa has learned more meaningful representation for understanding linguistic structures. Conversely, the advantage of the injected linguistic structures could be somewhat offset by the more sophisticated pre-training methodology employed by DeBERTa. Lastly, we note that while Mao et al. (2022) reported that their method (UniPELT) achieved significantly better performance compared to standard fine-tuning[1], our experiments with RoBERTa yielded the opposite conclusion[2]. This is consistent with the findings by Chen et al. (2022), where they find that PELT methods are highly unstable and cannot achieve consistently competitive performance compared to fine-tuning (especially in medium- to high-resource settings). Therefore, we hypothesize the discrepancy between our results and theirs is due to the likely extensive hyperparameter search conducted by (Mao et al., 2022), whereas we used the identical hyperparameter settings across all experiments as reported in subsection 5.4.

In Table 3, we report the number of trainable parameters for each of the methods in Table 2. While increasing the number of RGCN modules or hidden layer dimensions could improve the performance of our model, our hyperparameters settings (subsection 5.4) are selected specifically to match the number of parameters used in UniPELT (Mao et al., 2022). Additionally, it is worth mentioning that we elected not to incorporate the dependency relations since the number of parameters in RGCN layers increases linearly with the number of relation types.

| Method | Parameters |
|---|---|
| Fine-tuning | 110M (100%) |
| Adapter | 895K (0.81%) |
| Prefix-tuning | 184K (0.17%) |
| LoRA | 295K (0.27%) |
| UniPELT (AP) | 1.1M (0.99%) |
| UniPELT (APL) | 1.4M (1.26%) |
| Ours | 1.2M (1.14%) |

Table 3: Number of trainable parameters required for each parameter-efficient fine-tuning method.

## 6 Analysis

In this section, we provide an analysis of the model's behavior by first examining the linguistic expert used for each model (§6.1) before examining the convergence rate of Gumbel-Softmax gates at different layers of the model (§6.2).

### 6.1 Gate Values

Figure 2 illustrates the experts used at each layer of the models. At first glance, we can clearly see that all models tend to favor RGCN modules at the upper layers, while the standard MLP adapter is used for lower layers. This could be due to

---

[1]Since the original BERT paper (Devlin et al., 2019) does not report the GLUE development set results, the full fine-tuning results for BERT in Table 2 are copied from Mao et al. (2022).

[2]The full fine-tuning results for RoBERTa and DeBERTaV3 are copied for their original papers (Liu et al., 2019; He et al., 2021), where He et al. (2021) only reported the full set of GLUE results for their large variant. In both papers, the reported results are limited to three significant digits.

the fact that pre-trained language models are designed to learn hierarchical representations of the input, where lower-level layers typically capture the surface knowledge required to understand high-order compositions (Tenney et al., 2019a; Niu et al., 2022). Since such knowledge are generally applicable to all downstream tasks with minimal modification even during full fine-tuning (Zhou and Srikumar, 2022), augmenting their representations with compositional structures could be detrimental to the performance. Similar findings have also been reported by Rücklé et al. (2021), where dropping out adapters in the lower layers has the least amount of impact on model performance.

To gain a better understanding of how different linguistic structures are utilized, we provide a qualitative comparison of linguistic experts used between models. From Figure 2, we can see that the experts used between BERT and RoBERTa are very similar, with 5 out of the 8 tasks being exactly the same. In contrast, DeBERTa tends to use more semantic and syntactic experts, with no sequential experts selected on the top layer. We believe this is due to the disentangled attention mechanism used by the DeBERTa model (He et al., 2021), where the token positions are already encoded by an additional vector at each layer of the model. Additionally, we see that semantic graphs are selected the least. This could be due to the fact that we do not model the relation types, which are necessary to determine the nuanced semantic roles between concepts and ideas. Conversely, the relation types in syntactic trees (e.g., subject-verb) do not provide the full meaning of the sentence beyond grammatical structure, where prior studies have shown that syntax trees with no relations can still be beneficial for downstream performance (Bai et al., 2021).

## 6.2 Gate Convergence

Next, we examine the convergence rate for the gate logits by measuring the changes in the gate value between steps. For the purpose of analysis, we train the full set of experts for 2000 steps while keeping all hyperparameters the same. Figure 3 plots the JS-Divergence between the gate values' softmax distribution in 10-step intervals. From the plot, we can see that the gate values in the lower layers change rapidly in the early iterations before converging. This implies that the model can quickly learn to select the MLP module (§6.1), providing further evidence against injecting structural knowl-

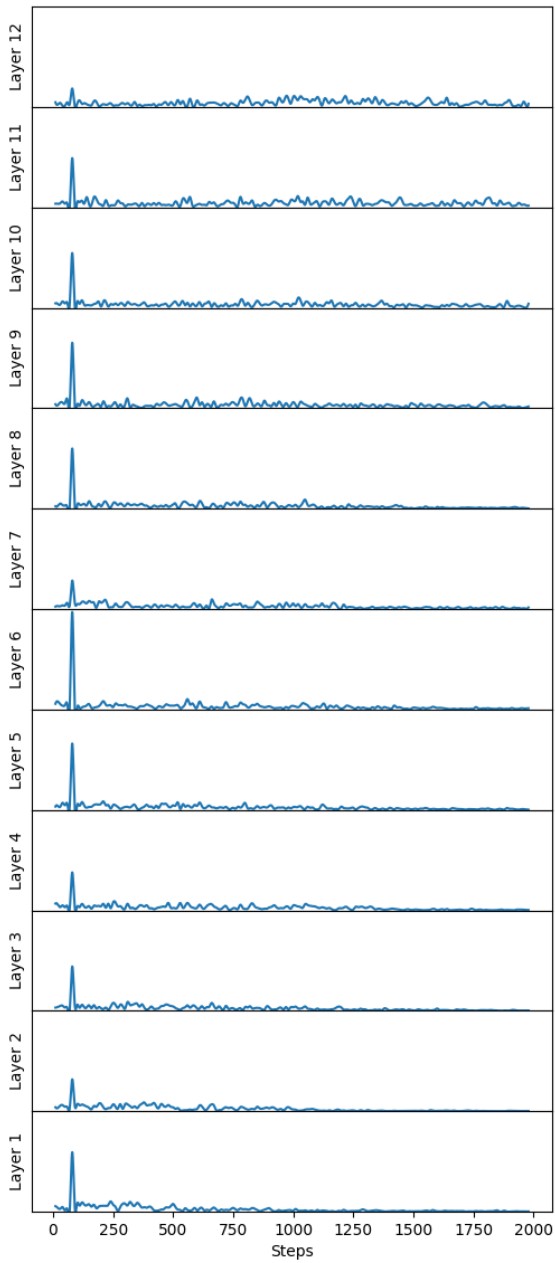

Figure 3: Average JS-Divergence between the gate value distribution, measured in 10-step intervals.

edge at lower layers of pre-trained models. In the upper layers, while it follows a similar trend where the gates change quickly before the change curve flattens out, we still see a moderate amount of oscillation even after 1000 steps. This can be interpreted as the best expert not having enough relative advantage over the others for the model to assign a high importance score. Since the main purpose of our work is to propose an architecture for selecting experts, we leave the in-depth investigation regarding the trade-off between different linguistic experts as an interesting venue for future work. Finally,

we see that almost all gates have converged at the 250-step mark. For reference, this is roughly 2% of the number of steps for a single epoch on the MNLI training set. This finding demonstrates that only a small number of steps are required to learn the importance of different adapter modules.

## 6.3 Ablation Study

We perform ablation experiments to study the effectiveness of our expert selection method based on the importance scores (section 4). To ensure a fair comparison with the results in Table 2, we only use one expert per layer while using the same architecture. We manually design the ordering of experts based on the intuition of the traditional NLP pipeline (Tenney et al., 2019a), with surface features at the bottom, syntactic features in the middle, and semantic features at the top (Jawahar et al., 2019). Specifically, we use sequential-graph encoding position information at the lower four layers, syntactic trees at the middle four layers, and semantic graph at the upper four layers. We also perform experience using only one expert for the entire model as a baseline.

| Method | SST-2 | QNLI |
|---|---|---|
| Syntax-only | 92.04 | 86.40 |
| Semantic-Only | 85.36 | 85.36 |
| Positional-Only | 88.97 | 88.97 |
| Manually-Designed | 91.84 | 89.12 |
| Ours | **94.27** | **92.53** |

Table 4: Ablation results with RoBERTa with manually selected experts, averaged across 3 seeds.

Table 4 shows the results for RoBERTa on the two medium-resource datasets (SST-2 and QNLI). From the results, we see that while our manually-designed approach achieved better performance than the single-expert models, they still significantly trail behind our automatic selection approach. This finding verifies our hypothesis that augmenting the representations of lower layers with compositional structures can have unrecoverable effects on the upper-layer representations used for task prediction (subsection 6.1), ultimately leading to a significant deterioration in performance.

## 7 Conclusion and Future Work

In this work, we introduce an approach that combines the two popular research areas of injecting linguistic structures and parameter-efficient fine-tuning (PEFT). To start, we introduce a novel framework that combines multiple linguistic structures in an architecture inspired by the Mixture-of-Experts model, where Gumbel-Softmax gates are used to learn the importance of these experts at different layers of the model in a small fixed number of training steps. Finally, we reduce the parameter count by pruning all but one expert at each layer such that the resulting number of trainable parameters is comparable to state-of-the-art PEFT methods. After running experiments with three different pre-trained models on the GLUE benchmark, the results show that our method can achieve the best overall performance while significantly outperforming the baselines on high-resource tasks. Finally, we examine the experts selected by each model and the convergence rate of the Gumbel-Softmax gates to gain a better understanding of the models' behavior and provide valuable insights for future studies on knowledge injection.

For future work, we plan to perform further experiments to determine the relative advantage of different linguistic knowledge and study how the quality of graphs affects model performance on downstream tasks. One significant challenge is to efficiently incorporate relation types of dependency trees, which we will explore in future work. In addition, we plan to further improve the efficiency of our approach by incorporating findings from other recent works, such as dropping adapters in the lower layers (Rücklé et al., 2021). Lastly, we plan to extend our approach to inject linguistic structures (including discourse graphs) into decoder-only architectures (Radford et al., 2019; Brown et al., 2020b) and perform studies on larger model variants (Touvron et al., 2023).

## Limitations

One limitation of our study is that our approach (excluding sequential graphs) requires high-quality parsers to construct gold-standard syntactic and semantic trees. While our approach is generally applicable to all structures, our experiments focus on sentence-level linguistic graphs on the GLUE benchmark. Other structures such as discourse trees on multi-sentential tasks remain to be explored in future studies. Additionally, all our experiments are performed on standard variants of pre-trained encoder models, different behavior could be observed on larger or differently structured models, such as the decoder-only architecture.

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
