# OpenReview forum: "Mixture-of-Linguistic-Experts Adapters for Improving and Interpreting Pre-trained Language Models"
_EMNLP/2023/Conference — EMNLP 2023 Findings_

### Official Review · Reviewer_otyM · 2023-08-01

**Typos Grammar Style And Presentation Improvements:** 1. The abstract and conclusion have s…
**Soundness:** 2

**Excitement:**

2: Mediocre: This paper makes marginal contributions (vs non-contemporaneous work), so I would rather not see it in the conference.

**Missing References:**

No.

**Paper Topic And Main Contributions:**

This paper proposes a method that combines linguistic structure injection and parameter-efficient fine-tuning (PEFT) of pre-trained language models. The method leverage a group of parallel adapter modules to encode different linguistic structures via Mixture-of-Linguistic-Experts architecture. To cut off the parameter budget, the method prunes the experts based on their important scores. Results on BERT, RoBERTa, DeBERTaV3 show that the proposed method outperforms baselines on high-resourced datasets. Besides, the paper analyzes the expert selection at each layer to provide some insights.

**Questions For The Authors:**

Please refer to the comments in the previous section.

**Reasons To Accept:**

1. The idea of incorporating linguistic knowledge into PLMs using adapter structure is well-motivated and novel.
2. By involving an Edgeless Graph expert, the authors also study whether linguistic knowledge is necessary to PLMs. This is interesting as previous works usually assume linguistic knowledge is beneficial to PLMs - such assumption, however, is not verified. The analysis of expert selection partly shows that additional linguistic knowledge is only useful to higher layers in the PLM architecture.

**Reasons To Reject:**

1. The experiment results are not strong enough to support the effectiveness of the proposed method. Though the authors argue that the proposed method perform considerably better in the four tasks with the highest number of training examples, I don't think improvement less than 1% can be viewed as "considerably better". For this claim, I think the authors should at least report the standard deviation in Table 2 to exclude the influence of randomness.
2. About the experiment setup, I think the authors should compare the results of the proposed method with the results of fine-tuning the full PLM. Though I'm aware that the method falls in the PEFT setting, I think the paper should compare the full fine-tuning to see whether the additional linguistic knowledge is indeed beneficial to the PLM.
3. The ablation study is missing. The paper should do ablation study to the involved linguistic information sources and the proposed expert pruning component.

**Reproducibility:**

4: Could mostly reproduce the results, but there may be some variation because of sample variance or minor variations in their interpretation of the protocol or method.

**Reviewer Confidence:**

4: Quite sure. I tried to check the important points carefully. It's unlikely, though conceivable, that I missed something that should affect my ratings.

---

> ### Author Rebuttal · Authors · 2023-08-29
>
> Thank you for the detailed review and constructive criticisms. We would like to take this opportunity to address your concerns and questions.
>
> ---
>
> __1. Experiment Results and Randomness__
>
> The reviewer brings up a valid concern. However, we argue that the 0.5-2% improvements in the high-resource datasets (SST-2, QNLI, QQP, MNLI) are noteworthy, especially when compared with the smaller improvements obtained in prior work on integrating linguistic knowledge (e.g., [Bai et al., 2021](https://aclanthology.org/2021.eacl-main.262/), [Wu et al, 2021](https://aclanthology.org/2021.tacl-1.14/)).
>
> More importantly, we agree with the reviewer that the standard deviation and statistical significance of these results should be reported; Table 2 will be revised to include those. More specifically, on the four high-resource datasets (SST-2, QNLI, QQP, and MNLI), the results from our model were significantly better than UniPELT (APL) with a confidence level of 99% according to the Bootstrap Significance test. We will update Table 2 with the standard deviation row (see below) and reorder the columns of such table by the training set size so that it will be clearer that our methods deliver significant gains for higher-resource datasets. All of this will much better support the results’ discussion starting on line 440.
>
> | Dataset           | CoLA | RTE  | MRPC | STS-B | SST-2    | QNLI     | QQP      | MNLI     |
> |-------------------|------|------|------|-------|----------|----------|----------|----------|
> | Training Set Size | 1K   | 2.5K | 2.7K | 5.8K  | 67K      | 105k     | 363K     | 392k     |
> | Standard deviaton | 1.32 | 2.43 | 0.45 | 0.22  | **0.32** | **0.47** | **0.06** | **0.18** |
> |                   |      |      |      |       |          |          |          |          |
>
> As a side note, the UniPELT results for running the significance tests are obtained based on the [`adapter-transformers`](https://github.com/adapter-hub/adapter-transformers) library ([Pfeiffer et al., 2020](https://aclanthology.org/2020.emnlp-demos.7/)), since the original paper by [Mao et al., 2022](https://aclanthology.org/2022.acl-long.433/) did not report the standard deviation of results across random seeds.
>
> ---
>
> __2. Results for Full Fine-tuning__
>
> We thank the reviewer for suggesting to compare our proposed method with the results of fine-tuning the full PLM. This also led us to think about considering performing the standard fine-tuning with the full set of experts at each layer (without Gumbel-Softmax) to understand the benefits of linguistic experts in the standard fine-tuning setting. Even though we recognize that these experiments are computationally rather demanding, we will try to include corresponding results in the final version.
>
> ---
>
> __3. Ablation Studies__
>
> Since there are a total of $4^{12}$ combinations for a 12-layer model (four choices of experts at each layer), exhaustive ablation experiments across all combinations would be computationally impractical, which is precisely the motivation of our selection-based pruning approach.  However, we agree that using a naive approach to selecting experts (e.g., only one expert type per model) could be useful for establishing a baseline and we are currently testing such an approach and will report the results in the final version.
>
> ---
>
> We will take the reviewer's suggestion and revise the conclusion section to make it more concise and distinct from the abstract.

---

### Official Review · Reviewer_bbmx · 2023-08-06

**Soundness:** 4

**Excitement:**

4: Strong: This paper deepens the understanding of some phenomenon or lowers the barriers to an existing research direction.

**Paper Topic And Main Contributions:**

This paper proposes combining parameter-efficient finetuning with injecting linguistic structures into large language models to create “Mixture-of-Linguistic Expert Adapters”. The adapters encoding different linguistic structures are inserted parallel to the feed-forward network in a transformer layer, with a Gumbel-softmax gate weighing the importance of each module. After the training, the model is pruned by removing all but one expert to reduce the parameter count. The resulting model is then analyzed to gain insight into the gate convergence rate and discover which expert is chosen at which layer.

**Questions For The Authors:**

No questions, great work!


**Reasons To Accept:**

The paper innovatively combines two methods to adapt pretrained language models successfully, leading to an average improvement on results across tasks.

The injection of linguistic knowledge to along with adapters is important to share with the wider community.

The paper is well written, clear and easy to follow.


**Reasons To Reject:**

The methods have only been tested on English. Despite the lack of availability of parses for lower-resource languages, it would be nice to see if this method works just as well for other languages.

**Reproducibility:**

5: Could easily reproduce the results.

**Reviewer Confidence:**

5: Positive that my evaluation is correct. I read the paper very carefully and I am very familiar with related work.

---

> ### Author Rebuttal · Authors · 2023-08-29
>
> Thank you for the thorough examination and the positive feedback! We appreciate your acknowledgment regarding the impact of our work, we hope our findings will motivate future studies on integrating various types of knowledge in the parameter-efficient setting.
>
> ---
>
> __Lower-Resource Languages__
>
> We also believe that as a next step, it is important to evaluate our method on other lower-resource languages (e.g., Tagalog, Bengali). One follow-up work we intend to do is to examine the degree to which the quality of the parses affects the model performance. This might provide some initial evidence of our method’s performance on low-resource languages with lower-quality parsers (e.g., [Şahin et al., 2018](https://aclanthology.org/D18-1545/))

---

### Official Review · Reviewer_y8mp · 2023-08-08

**Typos Grammar Style And Presentation Improvements:** 014 “Important” … 105 “Importance” I …
**Soundness:** 4

**Excitement:**

3: Ambivalent: It has merits (e.g., it reports state-of-the-art results, the idea is nice), but there are key weaknesses (e.g., it describes incremental work), and it can significantly benefit from another round of revision. However, I won't object to accepting it if my co-reviewers champion it.

**Paper Topic And Main Contributions:**

This paper combines injecting linguistic structure with Parameter Efficient Fine-Tuning (PEFT). The method relies on multiple different parts of the broader literature, such as Mixture-of-Experts (MoE) and the Gumball-Softmax trick. Overall, the method gets better results (in general) on a variety of tasks. Three linguistic graphs and non-linear transformations are applied to encode sentence-level structures. From Figure 2, it appears that the MLP is useful in most lower layers of the network, with various graph structures in the upper layers.

**Questions For The Authors:**

- I see that Table 2 is averaged over 3 runs. However, what is the variance on the runs? Are the results statistically significant?
- Table 3 shows the number of parameters for different methods. However, it is unclear to me how fast the different methods are. What is the computational complexity for the methods?
- How does varying the temperature of the Gumbel-Softmax affect results?

**Reasons To Accept:**

- Gains over the majority of the baselines on a variety of tasks

**Reasons To Reject:**

Despite some of the analysis, it is still a bit hard to discern what improvements the model is getting from what parts. For certain (but closely related) architectures, different linguistic structures appear to matter more. Likewise, how much does the Gumbel-Softmax impact this?

**Reproducibility:**

4: Could mostly reproduce the results, but there may be some variation because of sample variance or minor variations in their interpretation of the protocol or method.

**Reviewer Confidence:**

3: Pretty sure, but there's a chance I missed something. Although I have a good feel for this area in general, I did not carefully check the paper's details, e.g., the math, experimental design, or novelty.

---

> ### Author Rebuttal · Authors · 2023-08-29
>
> Thank you for the detailed review and feedback on our paper. We appreciate your insights and suggestions. We provide responses below to the raised concerns and questions.
>
> ---
>
> __What Improvements From Which Part__
>
> We acknowledge the importance of determining exactly what type of knowledge can benefit which part of the model, pinpointing such contributions requires exhaustive analysis by training a new model from scratch for each combination of knowledge type and layer. This is precisely the problem that we tackle. By sampling the contribution from a stochastic gate, our method provides an estimation of the improvement gain in an efficient way. In other words, the results from Table 2 show the marginal improvements the model gets from replacing the two MLP adapters with the selected linguistic experts (Figure 2). We will include the results for the MLP-experts-only configuration as a baseline in our final version.
>
> ---
>
> __Impact of Gumbel-Softmax__
>
> Regarding the influence of Gumbel-Softmax, we observed on the development sets that employing the standard softmax invariably defaults to the MLP adaptor across layers due to its ease of learning, resulting in an architecture similar to the standard Adapter baseline. In the revised paper we will clarify this point; namely, that the Gumbel-Softmax is critical for exploring a diverse combination of options, rather than defaulting to the local minima.
>
> ---
>
> __Variance and Statistical Significance__
>
> We report the standard deviation of our method with the BERT encoder in the table below.
>
> | Dataset           | CoLA | RTE  | MRPC | STS-B | SST-2    | QNLI     | QQP      | MNLI     |
> |-------------------|------|------|------|-------|----------|----------|----------|----------|
> | Training Set Size | 1K   | 2.5K | 2.7K | 5.8K  | 67K      | 105k     | 363K     | 392k     |
> | Standard deviaton | 1.32 | 2.43 | 0.45 | 0.22  | **0.32** | **0.47** | **0.06** | **0.18** |
> |                   |      |      |      |       |          |          |          |          |
>
> On the four high-resource datasets (SST-2, QNLI, QQP, and MNLI), the results from our model were significantly better than UniPELT (APL) with a confidence level of 99% according to the Bootstrap Significance test. We will update Table 2 with the standard deviation row and reorder the columns of such table by the training set size so that it will be clearer that our methods deliver significant gains for higher-resource datasets. All this will much better support the results’ discussion starting on line 440.
>
> As a side note, the UniPELT results for running the significance tests are obtained based on the [`adapter-transformers`](https://github.com/adapter-hub/adapter-transformers) library ([Pfeiffer et al., 2020](https://aclanthology.org/2020.emnlp-demos.7/)), since the original paper by [Mao et al., 2022](https://aclanthology.org/2022.acl-long.433/) did not report the standard deviation of results across random seeds.
>
> ---
>
> __Computational Complexity__
>
> The graph convolution operation has a computational complexity $\mathcal{O}(|E| \cdot d_1 \cdot d_2) $, where $d_1$ and $d_2$ are respectively the input and output dimension of the layer, and $|E|$ is the number of edges between nodes. In addition, the complexity of self-loop adds $\mathcal{O}(|N| \cdot d_1 \cdot d_2)$, where $|N|$ is the number of nodes. In our implementation based on the [`dgl`](https://github.com/dmlc/dgl) library ([Wang et al., 2019](https://arxiv.org/abs/1909.01315)), inference on GPUs is 2 times slower than the standard linear layer, which has the same complexity as self-loop. We will emphasize these points in our final draft.
>
> ---
>
> __Effects on Varying Temperature__
>
> Since we employ temperature annealing, the start and end values of the temperature are selected based on the recommendation from the survey by [Huijben et. al, 2021](https://arxiv.org/abs/2110.01515), with the number of steps determined empirically based on changes in gate values (Sec 6.2). Nevertheless, we agree that the start/end value of the temperature hyperparameter and the decaying methods (e.g., linear, exponential) may have an effect on model behavior. Therefore, we are running some experiments to establish how sensitive the models are to these changes. Results should be available for inclusion in the final version of the paper.

---

### Meta-Review · Area_Chair_wugr · 2023-09-17

**Recommendation:** 4

**Metareview:**

The reviewers agree that the method proposed in the paper is interesting and novel. They find the paper well written and easy to follow. Two of the reviewers seem to be concerned by the lack of ablation tests/clear indication of what information has an impact on the results. The authors' answer indicates that they can provide at least a preliminary analysis of this, should the paper be accepted. The reviewers ask a few other questions related to methodology/technical details, which are successfully answered by the authors.

The scores of one of the reviewers seem to be rather low given the wording of their review. The questions and objections of the reviewer are addressed in the authors' rebuttal. The reviewer seems to be reasonably satisfied with the answer, so I assume the reviewer forgot to increase the scores.

Overall, the reviews (and partially the scores) indicate that this is a sound paper which contains some novel ideas. The discussion during the rebuttal period will enable the authors to address some of the problems identified in the original version.

---

### Decision · Program_Chairs · 2023-10-07

**Decision:**

Accept-Findings

**Comment:**

The reviewers agree that the method proposed in the paper is interesting and novel. They find the paper well written and easy to follow. Two of the reviewers seem to be concerned by the lack of ablation tests/clear indication of what information has an impact on the results. The authors' answer indicates that they can provide at least a preliminary analysis of this, should the paper be accepted. The reviewers ask a few other questions related to methodology/technical details, which are successfully answered by the authors.

The scores of one of the reviewers seem to be rather low given the wording of their review. The questions and objections of the reviewer are addressed in the authors' rebuttal. The reviewer seems to be reasonably satisfied with the answer, so I assume the reviewer forgot to increase the scores.

Overall, the reviews (and partially the scores) indicate that this is a sound paper which contains some novel ideas. The discussion during the rebuttal period will enable the authors to address some of the problems identified in the original version.